# Real-Valued Direct Position Determination of Quasi-Stationary Signals for Nested Arrays: Khatri–Rao Subspace and Unitary Transformation

**DOI:** 10.3390/s22114209

**Published:** 2022-05-31

**Authors:** Haowei Zeng, Heng Yue, Jinke Cao, Xiaofei Zhang

**Affiliations:** 1College of Electronic and Information Engineering, Nanjing University of Aeronautics and Astronautics, Nanjing 211106, China; yhssz44@nuaa.edu.cn (H.Y.); caojinke@nuaa.edu.cn (J.C.); zhangxiaofei@nuaa.edu.cn (X.Z.); 2Key Laboratory of Dynamic Cognitive System of Electromagnetic Spectrum Space, Ministry of Industry and Information Technology, Nanjing 211106, China

**Keywords:** quasi-stationary signals, direct position determination, nested array, Khatri–Rao subspace, subspace data fusion, dimension-reduced, unitary transformation

## Abstract

The features of quasi-stationary signals (QSS) are considered to be in a direct position determination (DPD) framework, and a real-valued DPD algorithm of QSS for nested arrays is proposed. By stacking the vectorization form of the signal’s covariance for different frames and further eliminating noise, a new noise-eliminated received signal matrix is obtained first. Then, the combination of the Khatri–Rao subspace method and subspace data fusion method was performed to form the cost function. High complexity can be reduced by matrix reconstruction, including the modification of the dimension-reduced matrix and unitary transformation. Ultimately, the advantage of lower complexity, compared with the previous algorithm, is verified by complexity analysis, and the superiority over the existing algorithms, in terms of the maximum number of identifiable sources, estimation accuracy, and resolution, are corroborated by some simulation results.

## 1. Introduction

Source localization technology is an essential part of many fields, including rescue operation, resource exploration, intelligent transportation, and underwater detection [1,2,3,4]. Initially, typical localization methods, such as time of arrival (TOA) [5,6], angle of arrival (AOA) [7,8], and frequency difference of arrival (FDOA) [9,10] are always performed in a two-step mechanism. The intermediate parameters containing information regarding the source position are estimated first, such that the source position can be determined by methods that are based on the geometric relationship between the parameters. Nevertheless, the two-step algorithm cannot perform optimally, due to the inevitable information loss between the two steps. Moreover, a sharp decline in accuracy will be caused if parameter matching errors occur in multiple source scenarios.

To circumvent these problems in the two-step framework, a new one, called direct position determination (DPD), is first proposed in [11]. As its name suggests, the DPD algorithm is performed by processing the raw received signals to determine the source position. Thus, it skips the step of intermediate parameter estimation and takes the correlation among different received signals into account. The research results in [11] show that much higher accuracy can be achieved by DPD algorithms, compared with two-step algorithms, especially under low signal-to-noise ratio (SNR) conditions.

As the information associated with the source position cannot be completely ignored in DPD, a series of DPD algorithms based on different information types have been proposed. The information regarding TOA and AOA was considered in [11], where the maximum likelihood (ML) estimator was established, and multidimensional search was required for the determination of the source position. Though superior performance can be obtained, this algorithm is impractical, in the case of multiple sources, because of its high complexity. To cope up with this, a decoupled algorithm was proposed in [12], and the alternating projection algorithm was adopted in [13]. Besides, the subspace data fusion (SDF) DPD algorithm was proposed in [14], which can handle the problems of multiple sources better than the ML algorithm. It was actually an extension of the multiple signal classification (MUSIC) [15] algorithm. Except for the information regarding TOA and AOA, the information of FDOA can be also considered in the DPD algorithm. In [16], a ML estimator that contained the time difference of arrival (TDOA) and FDOA was designed, and its high computation load was avoided by the particle filter method. Different from the situation of moving receivers in [16], a moving source was considered in [17]. The delay and Doppler information were exploited in [17], and a new multiple particle filter algorithm was proposed to cope with the difficulty of estimating multiple parameters.

The above DPD algorithms were all designed for general Gaussian signals, while some research has demonstrated that an improved accuracy can be achieved if the specific properties of the source signals are considered. In [18,19], the DPD algorithms for the orthogonal frequency division multiplexing (OFDM) signals were proposed, where the ML estimator was exploited, and their superiority over the general algorithms was verified. Besides, some attempts at the DPD algorithms for non-circular (NC) signals have also been made. An improved SDF DPD algorithm for NC signals was proposed in [20], and its complexity was reduced by devising a Newton-type iterative method. Hereafter, some sparse arrays, such as nested array (NA) [21] and coprime array [22], were employed to obtain a larger array aperture, higher degrees of freedom (DOF), and higher accuracy [23,24,25]. In addition to the OFDM and NC signals, the properties of cyclo-stationary signals [26] can also be considered in the DPD algorithm.

Signals in the real world are always nonstationary, but locally stationary, such as speech and audio signals. These types of signals are called quasi-stationary signals (QSS), and they have stable second-order statistical properties within a short period of time and differ from any other frame [27]. To the best of our knowledge, none of the existing literature on DPD algorithms has considered and exploited the features of QSS. In this paper, which was inspired by related research regarding the direction of arrival (DOA) for QSS [27,28,29], the SDF DPD algorithm of QSS (QSS-SDF-DPD) for NAs is derived first. Moreover, the dimension-reduced matrix in the Khatri–Rao subspace method [29] is modified, and the unitary transformation [30,31] is adopted, so that the real-valued QSS-SDF-DPD (R-QSS-SDF-DPD) algorithm with a lower computational burden is proposed. We summarize the following contributions of this paper:The features of QSS are considered in the DPD model for NAs, where the cost function is constructed by combining the Khatri–Rao subspace and SDF methods, and the QSS-SDF-DPD algorithm is derived.The original dimension-reduced matrix is modified, and the unitary transformation method is exploited for the purpose of releasing the computational burden; then, the R-QSS-SDF-DPD algorithm is proposed.Apart from the given Cramer-Rao bound (CRB), the complexity analysis, summary of advantages, superiority of the R-QSS-SDF-DPD algorithm in terms of computational complexity, maximum number of identifiable sources, localization accuracy, and sources resolution is confirmed by some simulation experiments.

The remaining parts of this paper are organized as follows. Section 2 presents the system model of QSS-DPD for NAs. The proposed algorithm, which contains the Khatri–Rao subspace method for NA and matrix reconstruction for complexity reduction, is derived in Section 3, and a summary is given at the end. Section 4 provides the CRB, complexity analysis, and advantages of the proposed algorithm. Then, some relevant simulation results are shown in Section 5. The last section draws some conclusions on the paper.

Notation: Throughout the paper, the upper-case bold character and upper-case one are used to represent a matrix and vector, respectively, and a variable is denoted with a lower-case character. · and · represent the operation of taking the magnitude and Euclidean norm of a vector, respectively. E·, vec·, diag·, blkdiag·, and Re· represent the operation of expectation, vectorization, diagonalization, block diagonalization, and taking the real part, respectively. ⊗ and ⊙ represent the operation of the Kronecker and Khatri–Rao products, respectively. ℂM×N and ℝM×N represent the complex number set and real number set with dimension M×N, respectively. The operation of inverse, conjugate, transpose, and conjugate transpose are represented by ·−1, ·∗, ·T, and ·H, respectively. arcsin· represents the operation of arcsine. In, Jn, 0N, and 1n represent the n×n identity matrix, row-flipped form of the n×n identity matrix, n×1 vector with all zeros, and n×1 vector with all ones.

## 2. System Model

Consider the two-dimension scenario presented in Figure 1, where the K (it is assumed to be known, as it can be estimated by some methods [32,33,34,35]) far-field narrowband uncorrelated sources are intercepted by N base stations, which are equipped with a NA. As the location of the base stations are known, assume they are located at $v_{n}=\left[x_{n}^{v},y_{n}^{v}\right]\;\left(n=1,2,\ldots ,N\right)$, and the sources are located at $q_{k}=\left[x_{k},y_{k}\right]$
$\left(k=1,2,\ldots ,K\right)$. The specific structure of the M element NA that is exploited in this scenario is shown in Figure 2, where the first and second levels consist of $M_{1}$ and $M_{2}$ elements ($M=M_{1}+M_{2}$), respectively. The place of all physical array elements can be included in a set Θ, given by [21]:(1)$$\Theta =\left\{d_{i}=id|i=0,1,\ldots ,M_{1}-1\right\}\cup \left\{d_{j}=j(M_{1}+1)d+M_{1}d|j=0,1,\ldots ,M_{2}-1\right\}$$
where d denotes the unit adjacent spacing.

Assume the kth source is impinging on the nth base station from $\theta_{n,k}$; then, the received signal vector intercepted by the nth base station at the $t\mathrm{th}\;\left(t=1,2,\ldots ,T\right)$ sampling time can be presented by [14]:(2)$$x_{n}\left(t\right)=\sum_{k=1}^{K}a_{n}\left(\theta_{n,k}\right)s_{n,k}\left(t\right)+n_{n}\left(t\right)$$
where $a_{n}\left(\theta_{n,k}\right)={\left[e^{-j2\pi d_{1}\sin\theta_{n,k}/\lambda },\ldots ,e^{-j2\pi d_{m}\sin\theta_{n,k}/\lambda },\ldots ,e^{-j2\pi d_{M}\sin\theta_{n,k}/\lambda }\right]}^{T}$ denotes the steering vector of $\theta_{n,k}=\arcsin\left(\left(x_{k}-x_{n}^{v}\right)/\left\|q_{k}-v_{n}\right\|\right)$, which is the DOA from the kth source to the nth base station, $d_{m}\in \Theta \;\left(m=1,2,\ldots ,M\right)$, λ is the signal wavelength, $s_{n,k}\left(t\right)$ represents the envelope of kth source incident on the nth base station, and $n_{l}\left(t\right)$ is the Gaussian white noise vector.

For the sake of derivation, rewrite Equation (2), in the form of matrix, as [13]:(3)Xn=AnSn+Nn
where:(4)Xn=xn1,xn2,…,xnT∈ℂM×T
(5)An=anθn,1,anθn,2,…,anθn,K∈ℂM×K
(6)Sn=sn,1,sn,2,⋯,sn,KT=sn,11sn,12⋯sn,1Tsn,21sn,22⋯sn,2T⋮⋮⋱⋮sn,K1sn,K2⋯sn,KT∈ℂK×T
(7)Nn=nn1,nn2,…,nnT∈ℂM×T

Considering that the source signals conform to the properties of QSS [36,37,38], we assume $s_{n,k}\;\left(k=1,2,\ldots ,K\right)$ contains F frames of signals, and each frame source signal $s_{n,k,f}\;\left(f=1,2,\ldots ,F\right)$ contains $T_{F}$ snapshots ($T=FT_{F}$); then, $s_{n,1,f}$ satisfies [27]:(8)$$E\left\{{\left|s_{n,k,f}\right|}^{2}\right\}=g_{n,k,f}$$

Then, the fth frame received signals of the nth base station, which can be expressed as:(9)$$\begin{array}{cc} X_{n,f} & =\left[x_{n}\left((f-1)T_{F}+1\right),x_{n}\left((f-1)T_{F}+2\right),\ldots ,x_{n}\left(fT_{f}\right)\right]\\ & =A_{n}S_{n,f}+N_{n,f} \end{array}$$
where $S_{n,f}={\left[s_{n,1,f},s_{n,2,f},\ldots ,s_{n,K,f}\right]}^{T}$ denotes the fth frame source signal matrix, and $N_{n,f}$ is the corresponding additive noise.

According to Equations (8) and (9), a local covariance matrix can be defined by [27]:(10)$$R_{n,f}=E\left\{X_{n,f}X_{n,f}^{H}\right\}=A_{n}\Lambda_{n,f}A_{n}^{H}+C_{n}$$
where $\Lambda_{n,f}=diag\left\{g_{n,1,f},g_{n,2,f},\ldots ,g_{n,K,f}\right\}$ denotes the local source covariance matrix of the fth frame, and $C_{n}$ is the spatial noise covariance.

According to Property 1 in [27], the vectorization of Rn,f in Equation (10) can be expressed as:(11)yn,f=vecRn,f=vecAnΛn,fAnH+Cn=An∗⊙Angn,f+vecCn
where gn,f=gn,1,f,gn,2,f,…,gn,K,fT∈ℝK×1 is a real-value vector consisting of the diagonal elements in Λn,f.

Interestingly, a new received signal vector yn,f is obtained, whose steering matrix is An∗⊙An and noise vector is vecCn. By stacking the vectorization forms of all local covariance matrices, the new received signal matrix Yn∈ℂM2×F is obtained:(12)Yn=yn,1,yn,2,…,yn,F=An∗⊙AnGn+vecCn1FT
where Gn=gn,1,gn,2,…,gn,F∈ℝK×F is the new signal matrix.

Based on Assumption A4 in [27], the covariance of the source signal is time-varying. Therefore, $G_{n}$ can be treated as a new incoherent source signal matrix with F snapshots. Note that this is different from the general case, where the virtualization of the received signal makes the signals coherent, and some decoherence operations should be performed. In contrast, that problem does not occur in this paper, and the advantages of high DOF and large aperture can be fully preserved. As shown in Figure 3, we compare the DOF of the general signal model (the spatial smoothing method in [21] is adopted to overcome the coherent signal problem that is caused by the virtualization) and QSS model in this paper. Obviously, the QSS model has approximately twice as many DOF as the general signal model.

Define an orthogonal projector matrix H⊥=IF−1/F1F1FT∈ℝF×F, when post-multiplying Yn by it, the unknown noise in Equation (12) can be eliminated, and the new noise-eliminated received signal matrix Y˜n∈ℂM2×F can be given by [27]:(13)Y˜n=YnH⊥=An∗⊙AnGnH⊥+vecCn1FTH⊥=An∗⊙AnGnH⊥+vecCn1FTIM−1/F1F1FT=An∗⊙AnGnH⊥

According to Assumption A5 in [27], $G_{n}$ is a full row-rank matrix if F≥K+1, which means $rank\left(G_{n}^{T}\right)=K$. Note that $H^{⊥}$ is non-singular, and $G_{n}H^{⊥}$ is a full row-rank matrix. The noise subspace can be obtained after performing singular value decomposition (SVD) on ${\tilde{Y}}_{n}$ or eigenvalue decomposition (EVD) on its covariance matrix, so that the SDF algorithm [14] can be exploited to estimate the source positions.

However, some direct processing on ${\tilde{Y}}_{n}$ is time-consuming. To avoid this problem, the R-QSS-SDF-DPD will be proposed in the next section.

## 3. The Proposed Algorithm

In this section, the derivation process of the QSS-SDF-DPD and its modified real-valued version (R-QSS-SDF-DPD) are given in detail, including the Khatri–Rao subspace method for NA and matrix reconstruction for complexity reduction.

### 3.1. Khatri–Rao Subspace Method for NA

For a uniform linear array (ULA) with L elements, the new steering matrix An∗⊙An can be characterized as [27]:(14)A∗⊙A=ΓB∈ℂL2×K
where:(15)Γ=0⋯010⋯00⋯001⋯0⋮⋱⋮⋮⋮⋱⋮0⋯000⋯10⋯100⋯00⋯010⋯0⋮⋱⋮⋮⋱⋮⋮0⋯00⋯10⋮⋮⋮⋮⋮⋮⋮10⋯00⋯001⋯00⋯0⋮⋮⋱⋮⋮⋱⋮00⋯10⋯0∈ℝL2×2L−1
B=bθ1,bθ2,…,bθK∈ℂ2L−1×K denotes the dimension-reduced virtual array steering matrix, and bθk=ej2πL−1dsinθk/λ,…,ej2πdsinθk/λ,1,e−j2πdsinθk/λ,…,e−j2πL−1dsinθk/λT denotes the virtual steering vector of θk, which is the DOA of the kth source.

However, due to the discontinuity between the two-level elements of the NA, Equation (14) cannot be directly applied to it. To overcome this difficulty, a transformation should be applied to the original array steering matrix $A_{n}$.

For the two-level NA exploited in this paper, a total of 2RNA−1 RNA=M2M1+1 consecutive distinct elements can be obtained after the operation of Khatri–Rao product An∗⊙An. If we take the non-negative part of them as the virtual ULA, then the original NA can be treated as a subset of the virtual ULA, and their array steering matrices scarify [29]:(16)An=PB¯n
where P=pd1,pd2,…,pdMT∈ℝM×RNA is the selection matrix, and pdm∈ℝRNA×1 is a vector with 1 at the jth j=1+dm/d entry and 0 elsewhere [29]. B¯n=b¯n,1,b¯n,2,…,b¯n,K
∈ℂRNA×K represents the array steering matrix of the virtual ULA, and b¯n,k=1,e−j2πdsinθn,k/λ,…,e−j2πRNA−1dsinθn,k/λT∈ℂRNA×1 represents the corresponding steering vector of the kth source.

Substitute Equations (16) and (14) into Equation (13); then, Y˜n can be rewritten as [29]:(17)Y˜n=An∗⊙AnGnH⊥=PB¯n∗⊙PB¯nGnH⊥=P⊗PB¯n∗⊙B¯nGnH⊥=P⊗PΓ˜B˜nGnH⊥
where Γ˜∈ℝRNA2×2RNA−1 is the dimension-reduced matrix with the form of Equation (15), B˜n=b˜nθn,1,b˜nθn,2,…,b˜nθn,K∈ℂ2RNA−1×K denotes the dimension-reduced virtual array steering matrix with the form of B in Equation (14), and its kth column vector b˜nθn,k∈
ℂ2RNA−1×1 represents the corresponding steering vector with the form of bθk in B.

According to the definition of Γ˜ and P, they are both column orthogonal, so P⊗PΓ˜ is also column orthogonal, which is easy to verify [29]. Define V˜=P⊗PΓ˜TP⊗PΓ˜∈ℝ2RNA−1×2RNA−1, and the dimension of Y˜n can be reduced after a linear transformation, which is given by [29]:(18)Y⌣n=W˜Y˜n=V˜−1/2P⊗PΓ˜TY˜n=V˜−1/2P⊗PΓ˜TP⊗PΓ˜B˜nGnH⊥=V˜1/2B˜nGnH⊥∈ℂ2RNA−1×F
where Y⌣n is the dimension-reduced, noise-eliminated received signal matrix, and W˜=V˜−1/2P⊗PΓ˜T is the dimension-reduced matrix. As presented in Equation (18), the dimension of the original received signal matrix is reduced from RNA2 to 2RNA−1.

It can be seen from Equation (18) that a virtual noise-eliminated received signal matrix Y⌣n is obtained, whose array steering matrix is V˜1/2B˜n and source signal matrix is GnH⊥∈ℂK×F. Therefore, the covariance matrix of the dimension-reduced, noise-eliminated received signal matrix R⌣n∈ℂ2RNA−1×2RNA−1 can be calculated by:(19)R^n=Y⌣nY⌣nH/F

After the eigenvalue decomposition of ${\hat{R}}_{n}$, we obtain the corresponding signal subspace ${\hat{E}}_{n}^{S}$ and noise subspace ${\hat{E}}_{n}^{N}$, which are given by:(20)$${\hat{R}}_{n}={\hat{E}}_{n}^{S}{\hat{\Lambda }}_{n}^{S}{\left({\hat{E}}_{n}^{S}\right)}^{H}+{\hat{E}}_{n}^{N}{\hat{\Lambda }}_{n}^{N}{\left({\hat{E}}_{n}^{N}\right)}^{H}$$
where ${\hat{\Lambda }}_{n}^{S}=diag\left\{\sigma_{n,1}^{2},\sigma_{n,2}^{2},\ldots ,\sigma_{n,K}^{2}\right\}$ and ${\hat{\Lambda }}_{n}^{N}=diag\left\{\sigma_{n,K+1}^{2},\sigma_{n,K+2}^{2},\ldots ,\sigma_{n,2R_{NA}-1}^{2}\right\}$
$\left(\sigma_{n,1}^{2}>\sigma_{n,2}^{2},\right.$
$\left.\ldots ,\sigma_{n,K}^{2}>\sigma_{n,K+1}^{2}\geq \ldots \geq \sigma_{n,2R_{NA}-1}^{2}\right)$ denote the diagonal matrices made up of the maximum K eigenvalues and remaining ones, respectively. The corresponding eigenvectors form ${\hat{E}}_{n}^{S}$ and ${\hat{E}}_{n}^{N}$, respectively.

Hence, according to the SDF algorithm in [14], the spectrum function of the QSS-SDF-DPD algorithm for NAs spQSS−SDFp can be constructed by:(21)spQSS−SDFp=1∑n=1Nb˜nθpHV˜1/2E^nNE^nNHV˜1/2b˜nθp
where p is the position of search grid point in the search area Ξ, and b˜nθp∈ℂ2RNA−1×1 represents the steering vector of p for the nth base station, which has the same form as bθk in B. After finding the maximum K points of Equation (21), all source positions can be estimated.

### 3.2. Matrix Reconstruction for Complexity Reduction

Note that high complexity cannot be avoided during the peak search process of $sp_{SDF}\left(p\right)$, due to the existence of ${\tilde{V}}^{1/2}$ and lots of complex-valued multiplications. For the purpose of releasing the computational burden, the dimension-reduced matrix will be modified, and the unitary transformation method [30,31] will be adopted.

It can be observed from Equation (18) that, if we pre-multiply Y⌣n by V˜−1/2, we can obtain:(22)Y⌣nM=V˜−1/2Y⌣n=B˜nGnH⊥=V˜−1V˜B˜nGnH⊥=V˜−1P⊗PΓ˜TP⊗PΓ˜B˜nGnH⊥=V˜−1P⊗PΓ˜TY˜n=W˜MY˜n
where Y⌣nM is the modified, dimension-reduced, noise-eliminated received signal matrix, and W˜M=V˜−1P⊗PΓ˜T∈ℝ2RNA−1×M2 is the modified dimension-reduced matrix. 

Interestingly and coincidentally, for the NA, ${\tilde{W}}^{M}$ is equal to the Equation (9) in [39], which is expressed by:(23)$$\begin{array}{cc} {\left[{\tilde{W}}^{M}\right]}_{r,j+(k-1)M} & =\left\{\begin{array}{l} \frac{1}{\omega \left(\Delta_{j,k}\right)},\;\Delta_{j,k}=r-R_{NA}\\ 0,\;\mathrm{otherwise} \end{array}\right.\\ & r=1,2,\ldots ,2R_{NA}-1,j,k=1,2,\ldots ,M, \end{array}$$
where $\Delta_{j,k}=\left(d_{j}-d_{k}\right)/d\;\left(d_{j},d_{k}\in \Theta \right)$ represents the position difference between $d_{j}$ and $d_{k}$, and $\omega \left(\Delta_{j,k}\right)$ represents the number of pairs $\left(d_{j},d_{k}\right)$, whose difference $\left(d_{j}-d_{k}\right)/d$ is equal to $\Delta_{j,k}$.

According to Equation (22), Y⌣nM can be treated as a modified noise-eliminated received signal matrix, whose array steering matrix is B˜n and source signal matrix is GnH⊥∈ℝK×F. Note that, compared with the virtual array steering matrix V˜1/2B˜n before modification, V˜1/2 is eliminated, which means the matrix V˜1/2 would be removed from spSDFp. Thus, it partially reduces the computational burden.

As the virtual array configuration forming ${\tilde{B}}_{n}$ is centrosymmetric, the unitary transformation method [30,31] can be directly adopted.

Define the unitary matrix by [30]:(24)$$U_{2R_{NA}-1}=\frac{1}{\sqrt{2}}\left[\begin{array}{ccc} I_{R_{NA}-1} & 0_{R_{NA}-1} & jJ_{R_{NA}-1}\\ 0_{R_{NA}-1}^{T} & \sqrt{2} & 0_{R_{NA}-1}^{T}\\ J_{R_{NA}-1} & 0_{R_{NA}-1} & -jI_{R_{NA}-1} \end{array}\right]$$

Then, a real-valued matrix Y⌣nMR∈ℝ2RNA−1×F can be obtained by the unitary transformation given by:(25)Y⌣nMR=U2RNA−1HY⌣nM

Correspondingly, a real-valued covariance matrix R^nMR∈ℝ2RNA−1×2RNA−1 can be expressed by:(26)R^nMR=Y⌣nMRY⌣nMRH/F

Hence, after performing eigenvalue decomposition on ${\hat{R}}_{n}^{MR}$, we obtain:(27)$${\hat{R}}_{n}^{MR}={\hat{E}}_{n}^{MRS}{\hat{\Lambda }}_{n}^{RS}{\left({\hat{E}}_{n}^{MRS}\right)}^{H}+{\hat{E}}_{n}^{MRN}{\hat{\Lambda }}_{n}^{RN}{\left({\hat{E}}_{n}^{MRN}\right)}^{H}$$
where ${\hat{E}}_{n}^{MRS}$ is the real-valued signal subspace made up of eigenvectors corresponding to the largest K eigenvalues, ${\hat{E}}_{n}^{MRN}$ is the real-valued noise subspace made up of the remaining eigenvectors, and ${\left(\sigma_{n,1}^{R}\right)}^{2}>{\left(\sigma_{n,2}^{R}\right)}^{2},\ldots ,{\left(\sigma_{n,K}^{R}\right)}^{2}>{\left(\sigma_{n,K+1}^{R}\right)}^{2}\geq \ldots \geq {\left(\sigma_{n,2R_{NA}-1}^{R}\right)}^{2}$ are eigenvalues of ${\hat{R}}_{n}^{MR}$; the first K ones form the diagonal matrix ${\hat{\Lambda }}_{n}^{RS}$, and the last $2R_{NA}-K-1$ ones form the diagonal matrix ${\hat{\Lambda }}_{n}^{RN}$.

Finally, fuse all the real-value noise subspace of all base stations, and the spectrum function of R-QSS-SDF-DPD algorithm for NAs spR−QSS−SDFp can be expressed by:(28)spR−QSS−SDFp=1∑n=1Nb˜nRθpHE^nMRNE^nMRNHb˜nRθp
where b˜nRθp=U2RNA−1Hb˜nθp=2cos2πRNA−1dsinθp/λ,…, cos2πdsinθp/λ,1/2,sin2πdsinθp/λ,…,sin2πRNA−1dsinθp/λ∈ℝ2RNA−1×1 is the real-valued peak search steering vector. Then, find the maximum K points of spMR−QSS−SDFp, and take the corresponding coordinate positions as the estimates of all source positions.

### 3.3. Summary of The Proposed Algorithm

As a summary, the main steps of the QSS-SDF-DPD and R-QSS-SDF-DPD algorithms are listed below, where the first six steps belong to the former, and the last four ones belong to the latter.

Step 1: Calculate the covariance matrix of the received signal for each frame $R_{n,f}$ by Equation (10).

Step 2: Construct the vectorization forms of all $R_{n,f}$, and stack them together to obtain the new received signal matrix $Y_{n}$ by Equation (12).

Step 3: Eliminate the unknown noise, according to Equation (13).

Step 4: Obtain the original dimension-reduced, noise-eliminated received signal matrix ${\overset{⌣}{Y}}_{n}$ by Equation (18).

Step 5: Calculate the covariance matrix of ${\overset{⌣}{Y}}_{n}$ by Equation (19), and obtain the noise subspace matrix ${\hat{E}}_{n}^{N}$ by Equation (20).

Step 6: Construct the original spectrum function of QSS-SDF-DPD $sp_{QSS-SDF}\left(p\right)$ by Equation (21), and estimate all source positions by finding the maximum K points.

Step 7: Modify the dimension-reduced matrix to obtain the modified noise-eliminated received signal matrix ${\overset{⌣}{Y}}_{n}^{M}$ according to Equation (22).

Step 8: Obtain the real-valued matrix ${\overset{⌣}{Y}}_{n}^{MR}$, according to Equation (25).

Step 9: Calculate the real-valued covariance matrix ${\hat{R}}_{n}^{MR}$ by Equation (26), and obtain the real-valued noise subspace matrix ${\hat{E}}_{n}^{MRN}$ by Equation (27).

Step 10: Construct the spectrum function of R-QSS-SDF-DPD $sp_{R-QSS-SDF}\left(p\right)$ by Equation (28), and estimate all source positions by finding the maximum K points.

## 4. Performance Analysis

### 4.1. CRB

Based on the derivation results in [24,40], the CRB of DPD for NAs can be expressed by:(29)$$CRB\left(q\right)=\frac{\sigma^{2}}{2}{\left\{\sum_{t=1}^{T}Re\left[S_{t}^{H}D^{H}\Pi_{A}^{⊥}DS_{t}\right]\right\}}^{-1}$$
where $\sigma^{2}$ is the variance of the noise, and:(30)$$q={\left[q_{1},q_{2},\ldots ,q_{K}\right]}^{T}$$
(31)$$S_{t}=I_{2K}⊗s_{t}$$
(32)$$s_{t}={\left[s_{1,t}^{T},s_{2,t}^{T},\ldots ,s_{N,t}^{T}\right]}^{T}$$
(33)$$\Pi_{A}^{⊥}=I-A{\left(A^{H}A\right)}^{-1}A^{H}$$
(34)$$A=blkdiag\left\{A_{1},A_{2},\ldots ,A_{N}\right\}$$
(35)$$D=\left[\frac{\partial A}{\partial x_{1}},\frac{\partial A}{\partial y_{1}},\frac{\partial A}{\partial x_{2}},\frac{\partial A}{\partial y_{2}},\ldots ,\frac{\partial A}{\partial x_{K}},\frac{\partial A}{\partial y_{K}}\right]$$
Where $\frac{\partial A}{\partial x_{k}}$ and $\frac{\partial A}{\partial y_{k}}$ denote the partial derivatives of A, with respect to $x_{k}$ and $y_{k}$, respectively.

### 4.2. Complexity Analysis

In this subsection, the complexity of the SDF-DPD, QSS-SDF-DPD, and R-QSS-SDF-DPD algorithms are compared. For the sake of fairness, the two-level NA is employed in all algorithms, and the spatial smoothing method [21] is adopted in SDF-DPD algorithm. Moreover, we count the number of real-valued multiplication operations, instead of the complex-valued multiplication operations, which are equivalent to four real-valued multiplication operations. Table 1 presents the results of the comparison, and the corresponding intuitive form is depicted in Figure 4, where the search area is assumed to be divided into $N_{x}\times N_{y}$ grids, $M=M_{1}+M_{2}$, and $R_{NA}=M_{2}(M_{1}+1)$.

It can be concluded that the proposed R-QSS-SDF-DPD algorithm has a much lower complexity than the QSS-SDF-DPD algorithm, and it is slightly higher than the SDF-DPD algorithm, which confirms the effectiveness of the dimension-reduced matrix modification and unitary transformation. This means that the R-QSS-SDF-DPD algorithm is more practical in engineering applications.

### 4.3. Advantages

Due to the utilization of the QSS features, modification of the dimension-reduced matrix, and unitary transformation, the proposed R-QSS-SDF-DPD algorithm possesses the following advantages, when compared to the existing algorithms.

More sources can be estimated than the traditional SDF-DPD algorithm, even when $K>R_{NA}-1$ (K is the actual number of sources, and $R_{NA}-1$ is the maximum number of identifiable sources for two-level NA exploiting spatial smoothing method);Larger array aperture, lower localization error, and higher resolution can be obtained, compared to the SDF-DPD algorithm;Less computational burden than the QSS-SDF-DPD algorithm, before dimension-reduced matrix modification and unitary transformation.

## 5. Simulation Results

We compare the estimated performance of the SDF-DPD, QSS-SDF-DPD, and R-QSS-SDF-DPD algorithms by calculating the root mean square error (RMSE), as defined by:(36)$$\mathrm{RMSE}=\frac{1}{K}\sum_{k=1}^{K}\sqrt{\frac{1}{N_{E}}\sum_{ne=1}^{N_{E}}\left[{\left({\hat{x}}_{k,ne}-x_{k}\right)}^{2}+{\left({\hat{y}}_{k,ne}-y_{k}\right)}^{2}\right]}$$
where $N_{E}$ denotes the number of Monte Carlo simulation experiments, and $\left[{\hat{x}}_{k,ne},{\hat{y}}_{k,ne}\right]$ denotes the estimate of $\left[x_{k},y_{k}\right]$ in the neth simulation experiment. $N_{E}$ is set to be 1000 in all the following simulation experiments, unless a special statement is given.

Besides, in order to compare the resolution of algorithms, the resolving probability $P_{r}$ [41] is defined by:(37)$$P_{r}=\frac{N_{s}}{N_{E}}\times 100\%$$
where $N_{s}$ is the number of simulation experiments in which the two sources are successfully distinguished. In this section, two sources are placed parallel to the x-axis, one of them is fixed at $q_{1}=\left[x_{1},y_{1}\right]m$, while the abscissa of the other $q_{2}=\left[x_{1}+\Delta x,y_{2}\right]m$ is adjusted, where Δx denotes the distance of the two sources. For each simulation experiment, if the locations of two sources are estimated and satisfy $\left|{\hat{x}}_{1}-x_{1}\right|<\Delta x/2,\;\left|{\hat{x}}_{2}-x_{2}\right|<\Delta x/2$ (${\hat{x}}_{1}$ and ${\hat{x}}_{2}$ are the estimated abscissa of the two sources), then this simulation experiment passes; otherwise, it fails.

Simulation 1: To verify the feasibility of the proposed R-QSS-SDF-DPD algorithm, the scatter plot is shown in Figure 5. In this simulation, the number of sources is K=7, and they are located at $q_{1}=[-800,400]m$, $q_{2}=[-200,-600]m$, $q_{3}=[-400,1000]m$, $q_{4}=[700,1300]m$, $q_{5}=[1200,0]m$, $q_{6}=[1600,400]m$, and $q_{7}=[800,500]m$; the number of base stations is N=4, and they are located at $v_{1}=[0,2000]m$, $v_{2}=[-700,-1600]m$, $v_{3}=[1100,-1200]m$, and $v_{4}=[2000,2100]m$. The structure of NA is $M_{1}=2$, $M_{2}=2$, SNR is 5 dB, and the number of frames is F=200; each frame consists of $T_{F}=600$ snapshots, and the number of Monte Carlo simulation experiments is 500. As demonstrated in Figure 5, the proposed algorithm can estimate all source positions, even when $K=7>R_{NA}-1=5$, which proves the first advantage that was mentioned in Section 4.3. Besides, the feasibility of the R-QSS-SDF-DPD algorithm, in regard to estimate accuracy, is also confirmed.

Simulation 2: Figure 6 depicts the RMSE performance of algorithms versus SNR, where the R-QSS-SDF-DPD algorithm is compared to its version before matrix reconstruction (QSS-SDF-DPD). Here, we consider two conditions, where ULA and NA are deployed. The number of sources is set to K=2, $q_{1}=\left[0,390\right]m$, and $q_{2}=[250,560]m$; the number of base stations is N=3, $v_{1}=[-700,-300]m$, $v_{2}=[200,-500]m$, and $v_{3}=[600,-200]m$. The structure of array is M=4 for ULA, and $M_{1}=2$, $M_{2}=2$ for NA. The number of frames is F=80, each frame consists of $T_{F}=500$ snapshots, and the SNR varies from −6 dB to 18 dB. Figure 6 presents the results, in which the R-QSS-SDF-DPD algorithm performs almost the same as the QSS-SDF-DPD algorithm, but its complexity is much lower (as shown in Figure 4). It means the proposed algorithm is more efficient, and more suitable for practical scenarios.

Simulation 3: Figure 7 depicts the RMSE performance of algorithms versus SNR, where the R-QSS-SDF-DPD algorithm is compared with the SDF-DPD algorithm which ignores the features of QSS. In this simulation experiment, the number of sources is K=2, $q_{1}=[0,390]m$, and $q_{2}=[60,580]m$; the number of base stations is N=3, $v_{1}=[-700,-300]m$, $v_{2}=[200,-500]m$, and $v_{3}=[600,-200]m$. The structure of the array is M=4 for ULA and $M_{1}=2$, $M_{2}=2$ for NA; the number of frames is F=400, each frame consists of $T_{F}=300$ snapshots, and the SNR varies from −3 dB to 15 dB. The results, depicted in Figure 7, corroborate that the R-QSS-SDF-DPD algorithm outperforms the SDF-DPD algorithm, whether ULA or NA is deployed. In particular, the R-QSS-SDF-DPD algorithm for NAs is the closest to the CRB, which benefits from the utilization of the QSS features, though a slightly higher complexity has been brought (according to Figure 4).

Simulation 4: Different from the last simulation, the function of RMSE versus the number of frames F is adopted in this simulation experiment. Except for the SNR, which is set to be 5 dB and with F varying from 50 to 400, the other parameters are the same as in simulation 3. As presented in Figure 8, it can be found that all algorithms are positively affected by the number of frames, and the R-QSS-SDF-DPD algorithm always outperforms the SDF-DPD algorithm, which is consistent with the results in simulation 3.

Simulation 5: Figure 9 evaluates the resolving probability versus the distance between the two sources (Δx). The SNR is assumed to be 0 dB, and the number of frames is F=200, each frame consists of $T_{F}=600$ snapshots; source 1 is located at $q_{1}=[0,520]m$, and the other one is located at $q_{2}=[\Delta x,520]m$, where Δx varies from 30 m to 150 m. The other parameters are the same as in simulation 3. The result in Figure 9 demonstrates that the resolving probability of the R-QSS-SDF-DPD algorithm for NAs reaches 100% when Δx is only 50 m, while the SDF-DPD algorithm for NAs reaches 100% when Δx is 90 m. Obviously, the advantage of the R-QSS-SDF-DPD algorithm, in terms of sources resolution, has been verified.

## 6. Conclusions

In this paper, the R-QSS-SDF-DPD algorithm for NAs is proposed. According to the features of QSS, the new noise-eliminated received signal matrix can be obtained by stacking the vectorized form of the signal covariance matrix for each frame and eliminating the unknown noise. Then, the Khatri–Rao subspace method is adopted to reduce the dimension of the cost function. Thereafter, the dimension-reduced matrix is modified, and the unitary transformation method is employed to release the computational burden. It has been verified by some theorical analysis and simulations that the R-QSS-SDF-DPD algorithm owns a lower complexity than the QSS-SDF-DPD algorithm and performs better than the general SDF-DPD algorithm, in terms of the maximum number of identifiable sources, localization accuracy, and source resolutions.

## Figures and Tables

**Figure 1 sensors-22-04209-f001:**
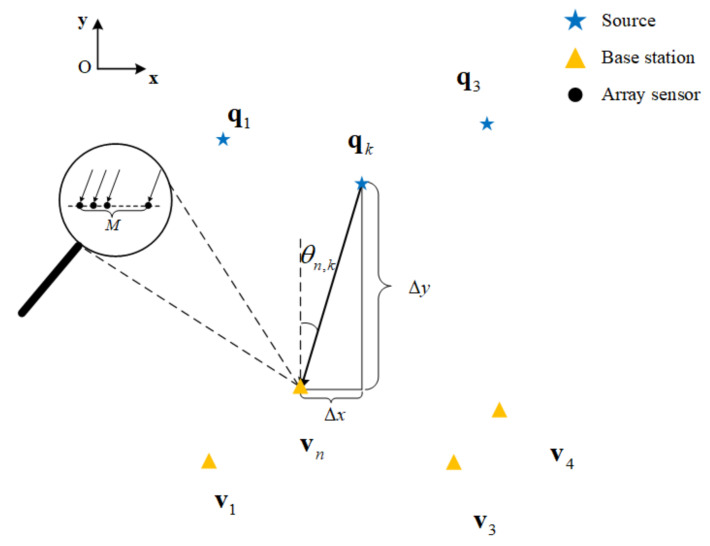
The localization scenario for multiple sources with multiple NAs.

**Figure 2 sensors-22-04209-f002:**
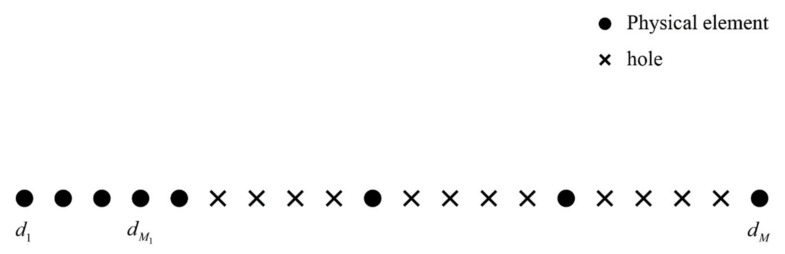
The structure of NA.

**Figure 3 sensors-22-04209-f003:**
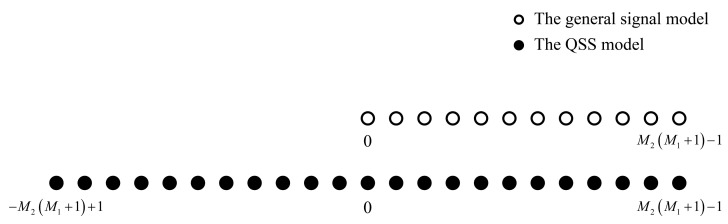
The DOF of the general signal model and QSS model.

**Figure 4 sensors-22-04209-f004:**
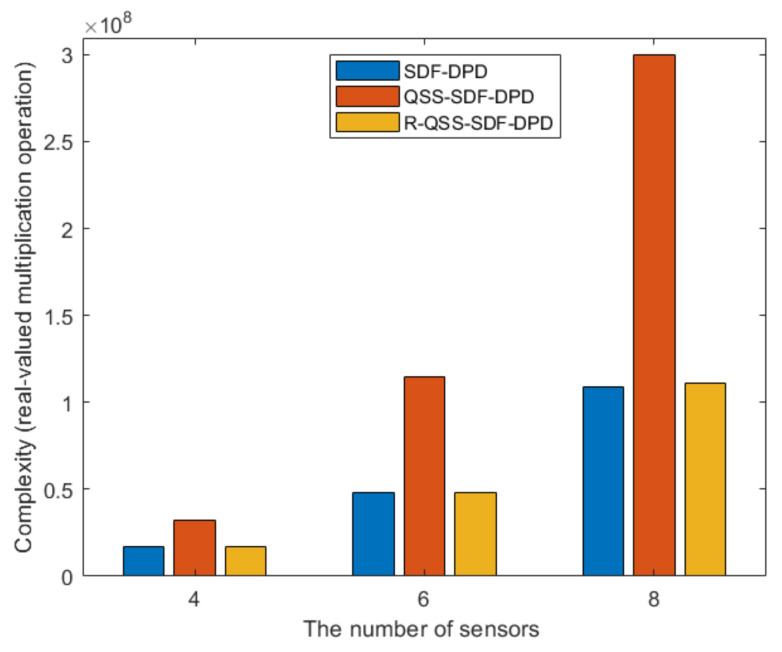
Comparison of complexity.

**Figure 5 sensors-22-04209-f005:**
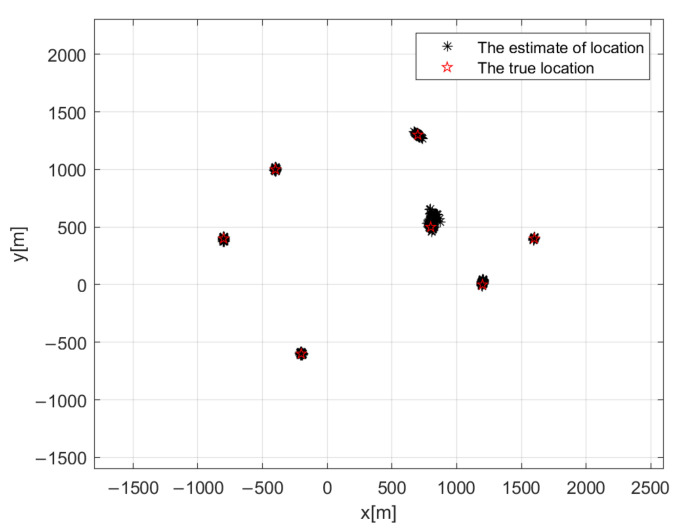
Scatter plots of the R-QSS-SDF-DPD algorithm for NAs.

**Figure 6 sensors-22-04209-f006:**
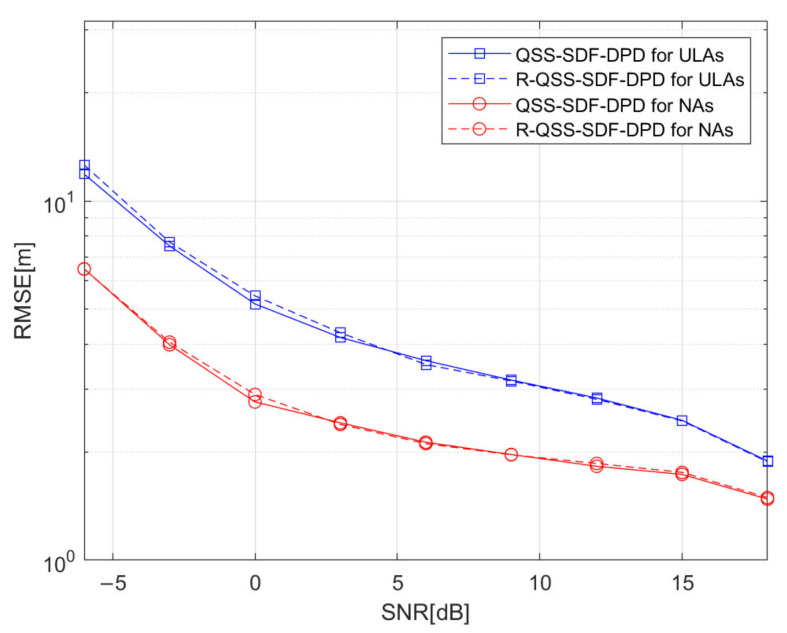
Comparison of QSS-SDF-DPD and R-QSS-SDF-DPD algorithms by RMSE versus SNR.

**Figure 7 sensors-22-04209-f007:**
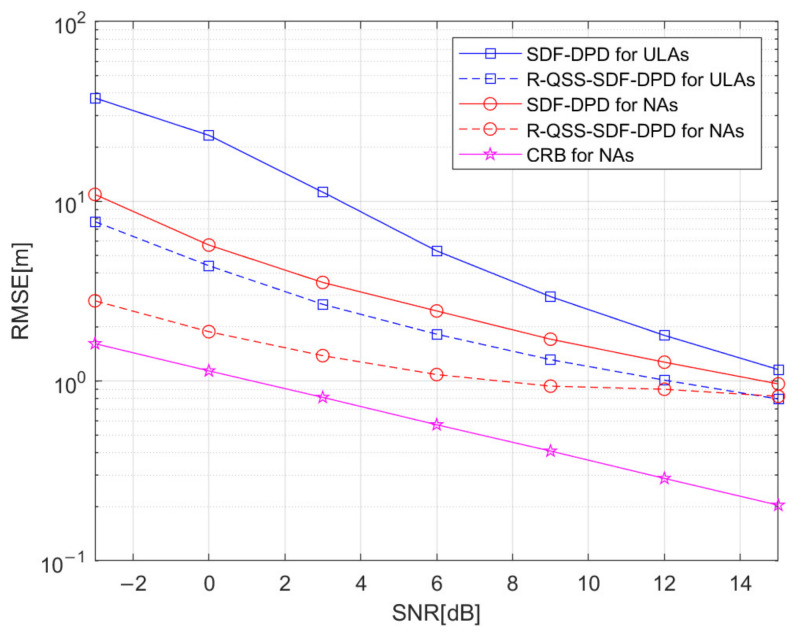
Comparison of SDF-DPD and R-QSS-SDF-DPD algorithms by RMSE versus SNR.

**Figure 8 sensors-22-04209-f008:**
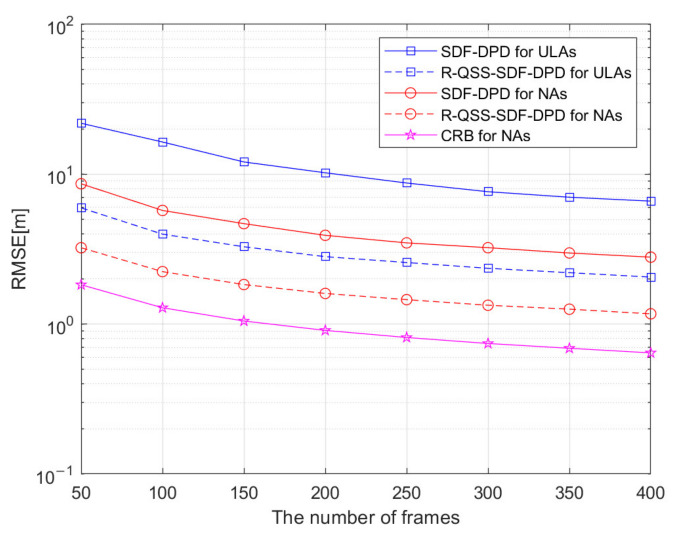
Comparison of SDF-DPD and R-QSS-SDF-DPD algorithms by RMSE versus the number of frames.

**Figure 9 sensors-22-04209-f009:**
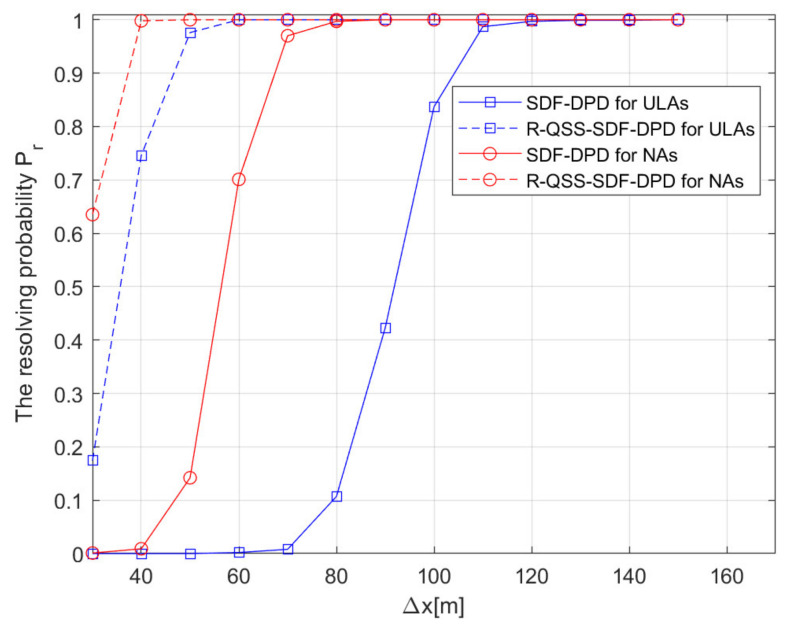
Comparison of SDF-DPD and R-QSS-SDF-DPD algorithms by resolving probability versus Δx.

**Table 1 sensors-22-04209-t001:** Complexity of algorithms.

Algorithm	Major Steps	Complexity (Real-Valued Multiplication Operation)
The SDF-DPD	Obtain covariance matrix	$O\{4NM^{2}T\}$
	Spatial smoothing	$O\{4NR_{NA}^{3}\}$
	EVD	$O\{4NR_{NA}^{3}\}$
	Spectral peak search	$O\{4N(R_{NA}^{2}(R_{NA}-K)+N_{x}N_{y}R_{NA}(R_{NA}+1))\}$
	Total	$O\{4N(M^{2}T+3R_{NA}^{3}-KR_{NA}^{2}+N_{x}N_{y}R_{NA}(R_{NA}+1))\}$
The proposed QSS-SDF-DPD	Obtain $R_{n,f}$	$O\{4NM^{2}T\}$
	Obtain ${\tilde{Y}}_{n}$	$O\{2NM^{2}F^{2}\}$
	Obtain ${\overset{⌣}{Y}}_{n}$	$O\{2NM^{2}F(2R_{NA}-1)\}$
	Obtain ${\hat{R}}_{n}$	$O\{4NF{\left(2R_{NA}-1\right)}^{2}\}$
	EVD	$O\{4N{\left(2R_{NA}-1\right)}^{3}\}$
	Spectral peak search	$O\{2N(3{\left(2R_{NA}-1\right)}^{2}(2R_{NA}-K-1)+4N_{x}N_{y}R_{NA}(2R_{NA}-1))\}$
	Total	$O\{2N(2M^{2}T+M^{2}F^{2}+M^{2}F(2R_{NA}-1)+(2F-3K)$ ${\left(2R_{NA}-1\right)}^{2}+5{\left(2R_{NA}-1\right)}^{3}+4N_{x}N_{y}R_{NA}(2R_{NA}-1))\}$
The proposed R-QSS-SDF-DPD	Obtain $R_{n,f}$	$O\{4NM^{2}T\}$
	Obtain ${\tilde{Y}}_{n}$	$O\{2NM^{2}F^{2}\}$
	Obtain ${\overset{⌣}{Y}}_{n}^{M}$	$O\{2NM^{2}F(2R_{NA}-1)\}$
	Obtain ${\overset{⌣}{Y}}_{n}^{MR}$	$O\{4NF{\left(2R_{NA}-1\right)}^{2}\}$
	Obtain ${\hat{R}}_{n}^{MR}$	$O\{NF{\left(2R_{NA}-1\right)}^{2}\}$
	EVD	$O\{N{\left(2R_{NA}-1\right)}^{3}\}$
	Spectral peak search	$O\{N({\left(2R_{NA}-1\right)}^{2}(2R_{NA}-K-1)+2N_{x}N_{y}R_{NA}(2R_{NA}-1))\}$
	Total	$O\{N(4M^{2}T+2M^{2}F^{2}+2M^{2}F(2R_{NA}-1)+(5F-K)$ ${\left(2R_{NA}-1\right)}^{2}+2{\left(2R_{NA}-1\right)}^{3}+2N_{x}N_{y}R_{NA}(2R_{NA}-1))\}$

## Data Availability

Not applicable.

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
