# Peer review of "Real-Valued Direct Position Determination of Quasi-Stationary Signals for Nested Arrays: Khatri–Rao Subspace and Unitary Transformation"

_sensors, 2022, doi:10.3390/s22114209_

Round 1

Reviewer 1 Report

This paper considers the problem of direct position determination for quasi-stationary signals with nested arrays. The KR subspace and subspace data fusion methods are used to build the cost function. The computational complexity is reduced by matrix reconstruction. This paper is a combination of several existing works, such as DOA estimation for QSS [27-29], SDF DPD [14], and KR subspace method [30]. The signal model, algorithm derivation and the corresponding results do not provide more insights. Therefore, the novelties and contributions are very limited. The current form cannot be recommended.

The new features of using SDF-DPD and QSS are not written clearly. If the eq. 8 is the core property of QSS and be used in the following derivation, the advantages should be clarified.

How it is possible to obtain a noiseless received signal matrix from noised received measurements?

There are lots of grammar and writing mistakes in the paper. The authors should check the paper thoroughly and carefully. 

Author Response

Response to Reviewer 1 Comments

Point 1: This paper considers the problem of direct position determination for quasi-stationary signals with nested arrays. The KR subspace and subspace data fusion methods are used to build the cost function. The computational complexity is reduced by matrix reconstruction. This paper is a combination of several existing works, such as DOA estimation for QSS [27-29], SDF DPD [14], and KR subspace method [30]. The signal model, algorithm derivation and the corresponding results do not provide more insights. Therefore, the novelties and contributions are very limited. The current form cannot be recommended.

Response 1: Thank the reviewer for pointing out the problem.

There are some differences between those DOA estimation algorithms in [27-29] and the proposed algorithm. As the former only estimates the DOAs of QSS and the conversion from parameters to position is required to determine the sources positions, it still belongs to the two-step framework. However, we propose a DPD algorithm for QSS in this paper, which can directly determine the positions of QSS without estimating intermediate parameters in advance and performs better than the two-step algorithms for QSS.

The SDF DPD algorithm in [14] is designed for general signal model via ULAs, which does not contain the features of the source. In contrast, the features of QSS are considered into the proposed algorithm and the NAs are exploited, which make the DOF of virtual array in the proposed algorithm is much more than the one in SDF DPD algorithm (for an array with  elements, the former is  and the latter is ). Besides, we convert the complex calculations to real-valued calculations so that the cost function of the proposed algorithm is as efficient as the SDF DPD algorithm while performing better.

Different from the KR subspace method in [30], we have modified the dimension-reduced matrix. As can be seen from  in eq. 21 and  in eq. 28, the linear transformation matrix  has been removed. This improvement avoids some unnecessary calculations and also makes the cost function more concise.

In general, this paper is different from the combined form of several existing works. As the proposed algorithm belongs to DPD framework, the DOA estimation algorithms for QSS in [27-29] belongs to the two-step framework. The features of QSS are considered into the system model and the NAs are exploited so that its DOF of virtual array is much more than the one of SDF DPD algorithm in [14]. The dimension-reduced matrix in KR subspace method [30] has been modified in this paper and we construct a more concise and efficient cost function than the one of the simple combined form.

This paper mainly has three contributions, they are:

  • Since many signals in the real world are not always stationary, such as speech and audio signals, which can be modeled as QSS. None of the exsiting literature on DPD algorithms has taken the features of QSS into consideration, we introduce them to the DPD framework, solve the corresponding problems and establish the algorithm model.
  • Considering the linear transformation of the noise subspace contained in the cost function of [27] and [29] brings some extra calculations, we thus modify the dimension-reduced matrix to avoid them and present a more concise and clear cost function.
  • The virtual array configuration forming is centrosymmetric, which means that the data in  is also centrosymmetric. Thus, the unitary transformation method in [30] is adopted to reduce the computational complexity. The complexity analysis and corresponding simulation results verify that the proposed R-QSS-SDF-DPD algorithm perfoms better than the SDF-DPD algorithm but cost slightly higher complexity (real-valued multiplication operation).

Point 2: The new features of using SDF-DPD and QSS are not written clearly. If the eq. 8 is the core property of QSS and be used in the following derivation, the advantages should be clarified.

Response 2: Thank the reviewer for the advice. According to the Assumption A4) in [29], the covariance of the source signal is time-varying. As a consequence,  can be treated as a new incoherent source signal matrix with  snapshots, it’s different from the general case where the virtualization of the received signal makes the signals coherent and some common decoherence methods are performed at the expense of DOF and array aperture, such as spatial smoothing method and Toeplitz method. As shown in the figure, compared with the general signal model (the spatial smoothing method in [21] is adopted to overcome the problem of coherent signals caused by the virtualization), the advantages of the QSS model in DOF and aperture can be fully preserved.

Action 2:

  1. System Model

According to Equation (8) and Equation (9), a local covariance matrix can be defined by [27]

Interestingly, a new received signal vector  is obtained, whose steering matrix is  and noise vector is . By stacking the vectorization forms of all local covariance matrices, the new received signal matrix  is obtained

where  is the new signal matrix.

Based on the Assumption A4) in [27], the covariance of the source signal is time-varying. Therefore,  can be treated as a new incoherent source signal matrix with  snapshots. Note that this is different from the general case where the virtualization of the received signal makes the signals coherent and some decoherence operations should be performed. In contrast, this problem does not occur in this paper, and the advantages of high DOF and large aperture can be fully preserved. As shown in Figure 3, we compare the DOF of the general signal model and the QSS model in this paper (the spatial smoothing method in [21] is adopted to overcome the coherent signal problem caused by the virtualization). Obviously, the QSS model has approximately twice as many DOF as the general signal model.

  • The DOF of the general signal model and the QSS model.

Point 3: How it is possible to obtain a noiseless received signal matrix from noised received measurements?

Response 3: Thank the reviewer for pointing out the problem. A noiseless received signal matrix cannot be obtained from noised received measurements, but the noise can be eliminated according to its characteristics (see Equation 11 and Equation 13 in [27]). According to [27], the correct description should be ‘noise-eliminated received signal matrix’, and we have corrected it.

Action 3:

Abstract: The features of quasi-stationary signals (QSS) are considered in direct position determination (DPD) framework, and a real-valued DPD algorithm of QSS for nested arrays is proposed. By stacking the vectorization form of signals covariance for different frames and further eliminating noise, a new noise-eliminated received signal matrix is obtained first. Then the combination of Khatri–Rao subspace method and subspace data fusion method is performed to form the cost function. High complexity can be reduced by matrix reconstruction, including modification of dimension-reduced matrix and unitary transformation. Ultimately, the advantage of lower complexity compared with the previous algorithm is verified by complexity analysis, and the superiority over the existing algorithms in terms of the maximum number of identifiable sources, estimation accuracy and resolution are corroborated by some simulation results.

  1. System Model

Define an orthogonal projector matrix , and post-multiplying  by it, then the unknown noise in Equation (12) can be eliminated and the new noise-eliminated received signal matrix  can be given by [27]

  1. The Proposed Algorithm

3.1. Khatri–Rao Subspace Method for NA

According to the definition of  and , they are both column orthogonal, so  is also column orthogonal, which is easy to be verified [29]. Define the dimension-reduced matrix , and the dimension of  can be reduced after a linear transformation given by [29]

where  is the dimension-reduced noise-eliminated received signal matrix. As presented in Equation, the dimension of the original received signal matrix is reduced from  to .

It can be seen from the Equation that a virtual noise-eliminated received signal matrix  is obtained, whose array steering matrix is  and source signal matrix is . Therefore, the covariance matrix of the dimension-reduced noise-eliminated received signal matrix  can be calculated by

3.2. Matrix Reconstruction for Complexity Reduction

It can be observed from Equation , if we pre-multiply  by , we can obtain

where  is the modified dimension-reduced noise-eliminated received signal matrix,   is the modified dimension-reduced matrix.

Interestingly and coincidentally, for the NA,  is equal to the Equation (9) in [39], which is expressed by

where  represents the position difference between  and , and  represents the number of pairs  whose difference  is equal to .

According to Equation ,  can be treated as a modified noise-eliminated received signal matrix whose array steering matrix is  and source signal matrix is . Note that, compared with the virtual array steering matrix  before modification,  is eliminated, which means the matrix  would be removed from . It thus partially reduces the computational burden.

3.3. Summary of The Proposed Algorithm

As a summary, the main steps of QSS-SDF-DPD algorithm and R-QSS-SDF-DPD algorithm are listed below, where the first six steps belong to the former and the last four ones belong to the latter.

Step 1: Calculate the covariance matrix of the received signal for each frame  by Equation .

Step 2: Construct the vectorization forms of all  and stack them together to obtain the new received signal matrix  by Equation (12).

Step 3: Eliminate the unknown noise according to Equation (13).

Step 4: Obtain the original dimension-reduced noise-eliminated received signal matrix  by Equation (18).

Step 5:  Calculate the covariance matrix of  by Equation (19), and obtain the noise subspace matrix  by Equation (20).

Step 6: Construct the original spectrum function of QSS-SDF-DPD  by Equation (21), and estimate all source positions by finding the largest  maximum points of it.

Step 7:  Modify the dimension-reduced matrix to obtain the modified noise-eliminated received signal matrix  according to Equation (22).

Step 8: Obtain the real-valued matrix  according to Equation (25).

Step 9: Calculate the real-valued covariance matrix  by Equation (26), and obtain the real-valued noise subspace matrix  by Equation (27).

Step 10: Construct the spectrum function of R-QSS-SDF-DPD  by Equation (28), and estimate all source positions by finding the largest  maximum points of it.

  1. Conclusions

In this paper, the R-QSS-SDF-DPD algorithm for NAs is proposed. According to the features of QSS, the new noise-eliminated received signal matrix can be obtained by stacking the vectorized form of the signal covariance matrix for each frame and eliminating the unknown noise. Then the Khatri-Rao subspace method is adopted to reduce the dimension of the cost function. Thereafter, the dimension-reduced matrix is modified and the unitary transformation method is employed to release the computational burden. It has been verified by some theorical analysis and simulations that the R-QSS-SDF-DPD algorithm owns lower complexity than QSS-SDF-DPD algorithm, and performs better than the general SDF-DPD algorithm in terms of the maximum number of identifiable sources, localization accuracy and sources resolution.

Point 4: There are lots of grammar and writing mistakes in the paper. The authors should check the paper thoroughly and carefully.

Response 4: Thank the reviewer for the comment. We have checked the paper thoroughly and carefully, and corrected some mistakes.

Action 4:

  1. Introduction

Moreover, a sharp decline in accuracy will be caused if errors of parameter matching occur in multiple sources scenarios.

The research results in [11] show that much higher accuracy than the two-step algorithms can be achieved by DPD algorithms, especially under low signal-to-noise ratio (SNR) conditions.

As the information associated with source position cannot be completely ignored in DPD, a series of DPD algorithms based on different information types have been proposed forward.

The delay and Doppler information is exploited in [17], and a new multiple particle filter algorithm is proposed to cope up with the difficulty of estimating multiple parameters.

Hereafter, some sparse arrays, such as nested array (NA) [21] and coprime array [22], are employed to obtain larger array aperture, higher degrees of freedom (DOF) and accuracy [23-25]. In addition to the OFDM signals and NC signals, the properties of cyclo-stationary signals [26] can also be considered in DPD algorithm.

To our best of knowledge, none of the existing literature on DPD algorithm has considered and exploited the features of QSS.

  1. System Model

Consider a two-dimension scenario presented in Figure 1, where  (it is assumed to be known, as it can be estimated by some methods [32-35]) far-field narrowband uncorrelated sources are intercepted by  base stations which are equipped with a NA. As the location of base stations are known, assume they are located at , and the sources are located at  . The specific structure of the  element NA exploited in this scenario is shown in Figure 2, where the first level and the second one consist of  and  elements (), respectively. The place of all physical array elements can be included in a set  given by [21]

  1. The Proposed Algorithm

3.1. Khatri–Rao Subspace Method for NA

According to the definition of  and , they are both column orthogonal, so  is also column orthogonal, which is easy to be verified [29]. Define , and the dimension of  can be reduced after a linear transformation given by [29]

where  is the dimension-reduced noise-eliminated received signal matrix,  is the dimension-reduced matrix. As presented in Equation, the dimension of the original received signal matrix is reduced form  to .

3.2. Matrix Reconstruction for Complexity Reduction

Note that high complexity cannot be avoided during the peak search process of  due to the existence of  and lots of complex-valued multiplications.

  1. Simulation Results

Simulation 3: Figure 7 depicts the RMSE performance of algorithms versus SNR, where the R-QSS-SDF-DPD algorithm is compared with the SDF-DPD algorithm ignoring the features of QSS. In this simulation experiment, the number of sources is , , , the number of base stations is , , , , the structure of array is  for ULA, and ,  for NA, the number of frames is , each frame consists of  snapshots, and the SNR varies from -3dB to 15dB. The result depicted in Figure 7 corroborates that the R-QSS-SDF-DPD algorithm outperforms the SDF-DPD algorithm whether ULA or NA is deployed. In particular, the R-QSS-SDF-DPD algorithm for NAs is the closest to the CRB, which benefits from the utilization of the QSS features, though slightly higher complexity has been brought (according to Figure 3).

  • Comparison of SDF-DPD algorithm and R-QSS-SDF-DPD algorithm by RMSE versus SNR.

Simulation 4: Different from the last simulation 3, the function of RMSE versus the number of frames  is adopted in this simulation experiment. Expect for the SNR is set to be 5dB and  varies from 50 to 400, other parameters are the same as simulation 3. As presented in Figure 8, it can be found that all algorithms are positively affected by the number of frames and the R-QSS-SDF-DPD algorithm always outperforms the SDF-DPD algorithm, which is consistent with the results in simulation 3.

  • Comparison of SDF-DPD algorithm and R-QSS-SDF-DPD algorithm by RMSE versus the number of frames.

Simulation 5: Figure 9 evaluates the resolving probability versus the distance between two sources . The SNR is assumed to be 0dB, the number of frames is , each frame consists of  snapshots, the source 1 is located at , the other one is located at , where  varies from 30m to 150m, other parameters are the same as simulation 3. The result in Figure 9 demonstrates that the resolving probability of the R-QSS-SDF-DPD algorithm for NAs reaches 100% when is only 50m, while the SDF-DPD algorithm for NAs reaches 100% when  is 90m. Obviously, the advantage of the R-QSS-SDF-DPD algorithm in terms of sources resolution has been verified.

  • Comparison of SDF-DPD algorithm and R-QSS-SDF-DPD algorithm by resolving probability versus .

Reviewer 2 Report

A direct position determination (DPD) framework with nested arrays, under quasi-stationary signals (QSS) condition, is proposed. The Khatri–Rao subspace method is used to identify a significant number of signal sources, verified by simulations. The proposed approach improves over existing approach mainly in the reduction of computational load. The contents and methods are well presented, and the conclusions are consistent with the simulation results. The references are appropriate. Please check the following items to refine the article.

[1] Line 120: The element positions of second array do not fit eqn.(1), please double check.

[2] Line 188: Please briefly describe “the discontinuity between the two-level elements of the NA”.

[3] Line 197: Please specify the definition of “d_m” and how are they selected.

Reviewer 3 Report

Real-Valued Direct Position Determination of Quasi-Stationary Signals for Nested Arrays: Khatri-Rao Subspace and Unitary Transformation 
by: Haowei Zeng, Heng Yue, Jinke Cao and Xiaofei Zhang

Source localization technology is an essential part of many fields, including rescue operation, resource exploration, intelligent transportation and underwater detection. Typical localization methods are always performed in a two-step mechanism. The intermediate parameters containing information of source position are estimated first such that the source position can be determined by some methods based on geometric relationship between the parameters. 
In the article, the Authors considered the features of quasi-stationary signals in direct position determination framework, and proposed a real-valued DPD algorithm of QSS for nested arrays. By stacking the vectorization form of signals covariance for different frames and further eliminating noise, a new noiseless received signal matrix is obtained first. Then the combination of Khatri–Rao subspace method and subspace data fusion method is performed to form the cost function. 
The structure of the manuscript is considered and clear. In the introduction, the background and comprehensive review of the problem's literature were presented. The Authors present system model and the proposed algorithm mathematically. Results of the research (simulations) have been presented in graphic form. Conclusions, on the basis of the research, are clear. 

Following suggestions should be taken into consideration:
Table 1: First column contains algorithms, so there is not necessary to write "algorithm" in each row
I suggest to use the bracket for presentation usnits of the value (variable): in Figure 4 rather x [m] and y [m] than x/m and y/m
In Figures 5, 6: rather RMSE [m] and SNR [dB} than RMSE/m and SNR/dB
